# Controlling factor of incoming plate hydration at the north-western Pacific margin

Gou Fujie[1], Shuichi Kodaira[1], Yuka Kaiho[1], Yojiro Yamamoto [1], Tsutomu Takahashi[1], Seiichi Miura[1] & Tomoaki Yamada[2]

Hydration of the subducting oceanic plate determines the amount of water transported from Earth's surface into its interior, and plate bending-related faulting (bend faulting) just prior to subduction is considered to promote hydration. Bend faulting shows significant spatial variation, but its contribution to hydration is still poorly understood. Here we present the results of controlled-source seismic surveys around the junction of the Japan and Kuril trenches. We found structural changes caused by bend faulting before subduction differed distinctly between both trenches and were well correlated with plate hydration after subduction, suggesting the bend faulting controls spatial variations in plate hydration. Differences in bend faulting are closely related to the angle between the current trench and the ancient spreading ridge, and the hydration is more extensive where this trench-ridge angle is oblique in the study area. Thus, we propose this angle is a major factor controlling plate hydration.

[1] Japan Agency for Marine-Earth Science and Technology (JAMSTEC), 3173-25, Showa-machi, Kanazawa-ku, Yokohama, 236-0001 Kanagawa, Japan.
[2] Earthquake Research Institute, University of Tokyo, Yayoi 1-1-1, Bunkyo-ku, 113-0032 Tokyo, Japan. Correspondence and requests for materials should be addressed to G.F. (email: fujie@jamstec.go.jp)

As a part of global mantle convection, water is transported by subducting oceanic plates from Earth's surface into its interior, and the degree of hydration of the subducting oceanic plate determines the amount of transported water. In subduction zones, dehydration occurring within the subducting oceanic plate induces intra-slab intermediate-depth earthquakes, and the expelled water leads to arc magmatism, promotes metamorphism, lowers the slab temperature, and consequently affects interplate coupling[1,2]. Thus, the degree of hydration of the oceanic plate before subduction is key to understanding these subduction zone phenomena as well as the global water cycle.

The first-order control on oceanic plate hydration has previously been considered to be hydrothermal circulation and alteration at the spreading ridge. Recently, seismic and electromagnetic structure studies in subduction zones around the world have revealed that plate bending-related normal faulting near the subduction trench also promotes hydration of the oceanic plate[3-11]. Unlike hydrothermal circulation near the spreading ridge, which is confined to the oceanic crust, bend faulting near the subduction trench can potentially cut across the entire crust and reach to the upper mantle (e.g., the 1933 Sanriku earthquake[12,13]). Thus, if bend faults act as pathways for water penetration into the mantle, the amount of water transported by the oceanic plate might be much larger than previously thought[14,15].

Bend faulting shows remarkable variation among subduction trenches. However, the actual contribution of bend faulting and its spatial variations to plate hydration remains poorly understood because plate hydration is affected by various hydration and dehydration processes from the spreading ridge to the subduction trench and isolating the contribution of each process is not straightforward.

At the north-western Pacific margin off Japan, the old oceanic Pacific plate, which formed at 120–130 Ma at a fast-spreading ridge, is subducting beneath the north-eastern Japan arc at the junction of the Japan and Kuril trenches[16]. Although the same oceanic plate is subducting with a similar curvature at each trench, horst and graben structures, which are seafloor topographic features related to bend faulting, exhibit remarkable dif-ferences between the trenches (Fig. 1). In the Japan Trench, the throw of the bend faults is more than 800 m, and the fault spacing (horizontal distance between two horsts) is 10–15 km (Fig. 2c). In contrast, in the Kuril Trench, the throw is less than half (at most 400 m), and the spacing is much less (roughly 5 km) (Fig. 2d). These differences make this trench junction a good place to investigate the contribution of bend faulting and its spatial variations to plate hydration.

In this study, we show the seismic structural changes caused by bend faulting before subduction are well correlated with plate hydration after subduction. Differences in bend faulting between these two trenches are closely related to the angle between the current trench axis and the ancient spreading ridge axis. These findings indicate that the trench-ridge angle is a major controlling factor of plate hydration in these subduction zones.

## Results

**Tomographic imaging.** To reveal the structural changes caused by bend faulting, we conducted extensive seismic surveys in the outer trench areas of these subduction zones (Fig. 1). We deployed Ocean Bottom Seismometers (OBSs) at intervals of 6 km along each survey line, and fired a large airgun array at 200-m intervals for OBSs and at 50-m intervals for a 6-km-long, 444-channel hydrophone streamer. The data quality was good (Supplementary Figs. 1-3). We succeeded in obtaining P-wave velocity ($V_p$) and S-wave velocity ($V_s$) models by tomographic traveltime inversion of both OBS and MCS data (see Methods).

The resultant $V_p$ models show a simple layered oceanic plate structure in which the oceanic crust is 7 km thick (Fig. 3). The topmost mantle $V_p$ outside of the bend faulting area was ~8.5 km s$^{-1}$ in the Kuril Trench profile (A2) but ~7.9 km s$^{-1}$ in the Japan Trench profile (A3). This large difference is well explained by seismic anisotropy due the alignment of olivine crystals in response to mantle flow at the spreading-ridge axis[17,18] and is not relevant to the bend faulting. On the other hand, the reduction of $V_p$ and increase of the $V_p/V_s$ ratio in the vicinity of the trench axis are considered to be related to the bend faulting (Fig. 3, Supplementary Figs 4-8). A reduction of $V_p$ is widely observed in the outer trench area of subduction

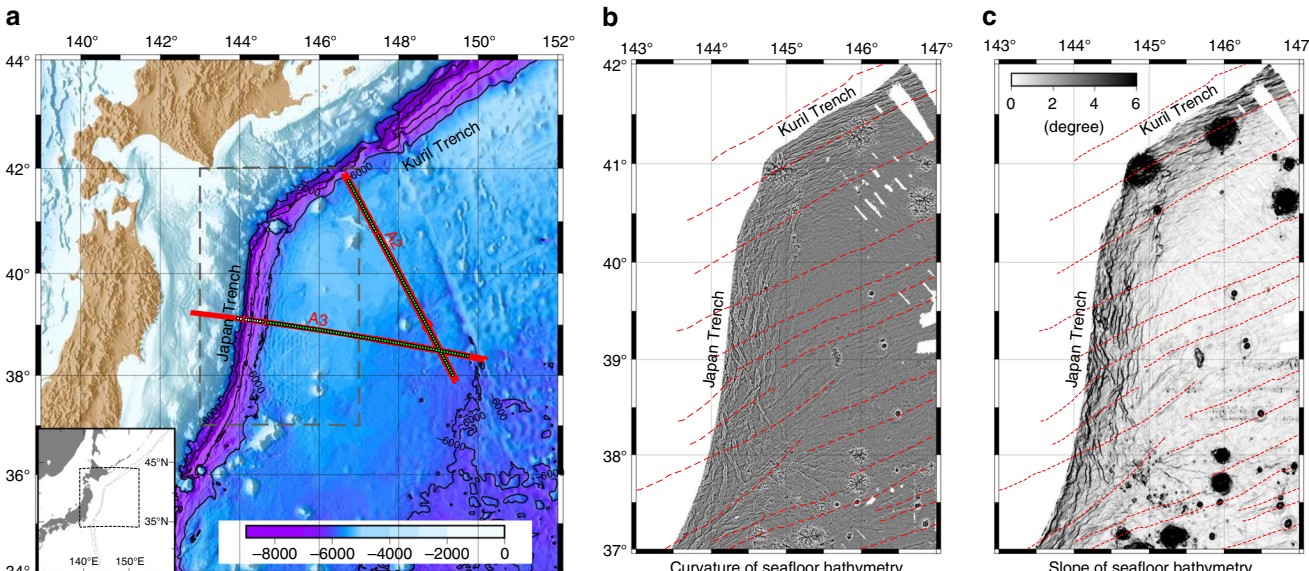

**Fig. 1** Study area. **a** Bathymetric map of the north-western Pacific margin off Japan around the junction of Kuril and Japan trenches (location in inset). Solid red lines A2 and A3 are seismic survey lines. White circles represent OBSs located in the vicinity of the Japan Trench axis, and green circles represent other OBSs. No OBSs were deployed in the vicinity of the Kuril Trench axis. **b** Curvature (Laplacian) of the seafloor topography in the outer trench area. **c** Slope gradient of the seafloor in the outer trench area. Dashed red lines in **b**, **c** are magnetic anomaly lineations[16]

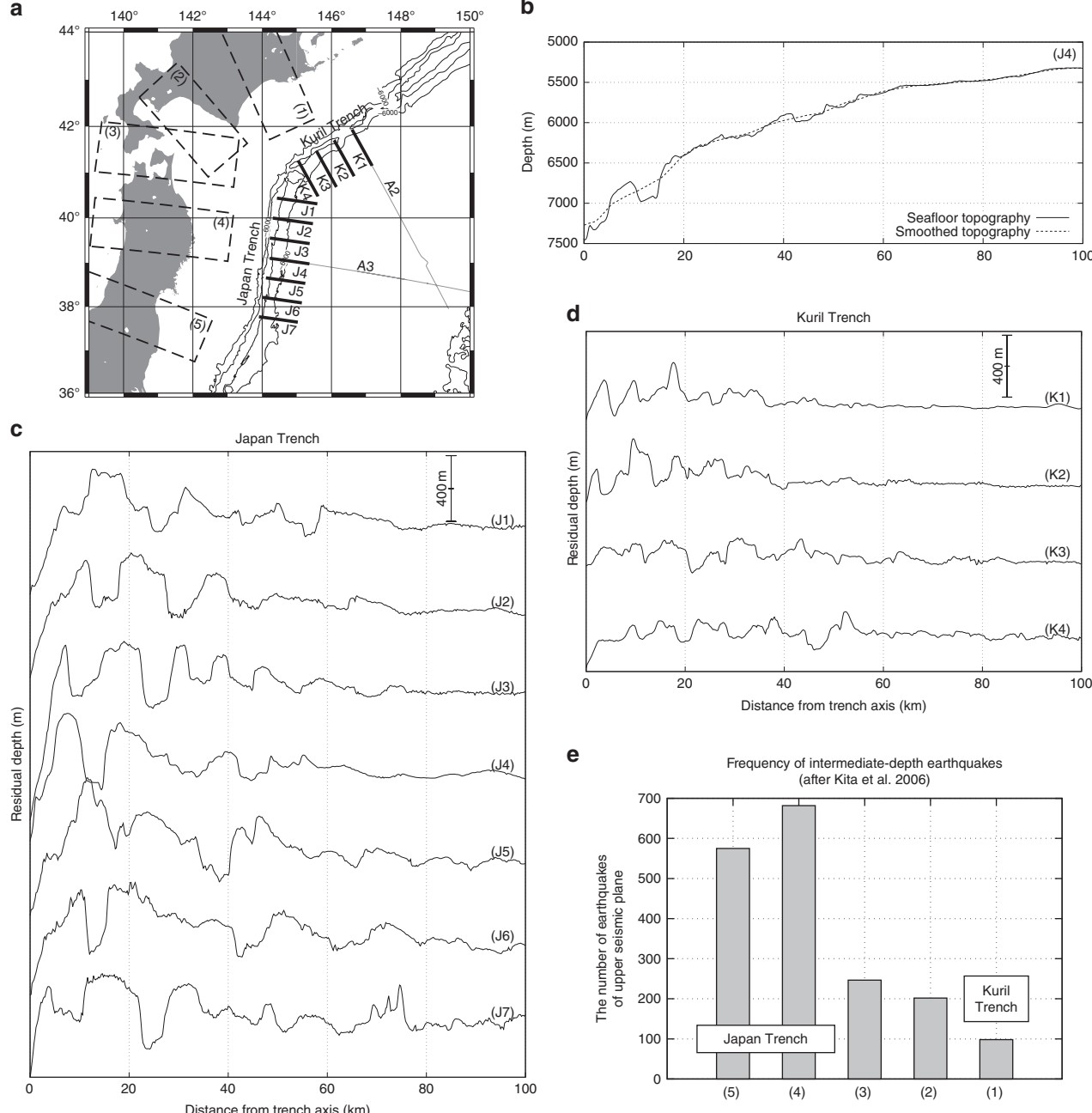

**Fig. 2** Seafloor topography. **a** Location map. Solid lines K1-K4 and J1-J7 are the locations of the seafloor topographic profiles shown in **b**–**d**, and boxes 1-5 are the areas for evaluating the frequency of the earthquakes in **e**. **b** Seafloor topography along profile J4: raw data (solid) and smoothed data (dashed). Smoothing was performed with a moving average spatial filter. **c** Seafloor topography after removing the trend, defined by difference between raw and smoothed data, in the Japan Trench. The horizontal axis is the distance from the trench axis. The throw of horst and graben reaches more than 800 m near the trench axis. **d** Seafloor topography after removing the trend in the Kuril Trench. The throw is <400 m. **e** Frequency of dehydration-induced intermediate-depth earthquakes after subduction[24]. The number of earthquakes was calculated for each box (1)–(5). The vertical axis shows the peak of the upper plane earthquake frequency. The activity of dehydration-induced earthquakes is remarkably higher in the Japan Trench than in the Kuril Trench

zones worldwide, where it is interpreted to be a consequence of fracturing, water penetration, and hydration[5,6,8,10,19]. A reduction of $V_P$ in the continental rifting area is similarly interpreted to be related with the water penetration and hydration caused by active faulting[20]. The increase of the $V_P/V_s$ ratio strongly suggests an increase in water content, because $V_s$ is more sensitive to the presence of water and hydrated minerals than $V_P$. Although we could not constrain the $V_P/V_s$ ratio within the mantle owing to heavy attenuation of S-wave refraction from

the mantle, the increase of the $V_P/V_s$ ratio within the crust indicates that the water content became higher in the bend faulting areas. Therefore, we considered the $V_P$ reduction to be accompanied by hydration as in other subduction zones and rifting areas.

**Spatial variations in bend faulting and plate hydration.** Structural changes related to bend faulting were commonly observed in

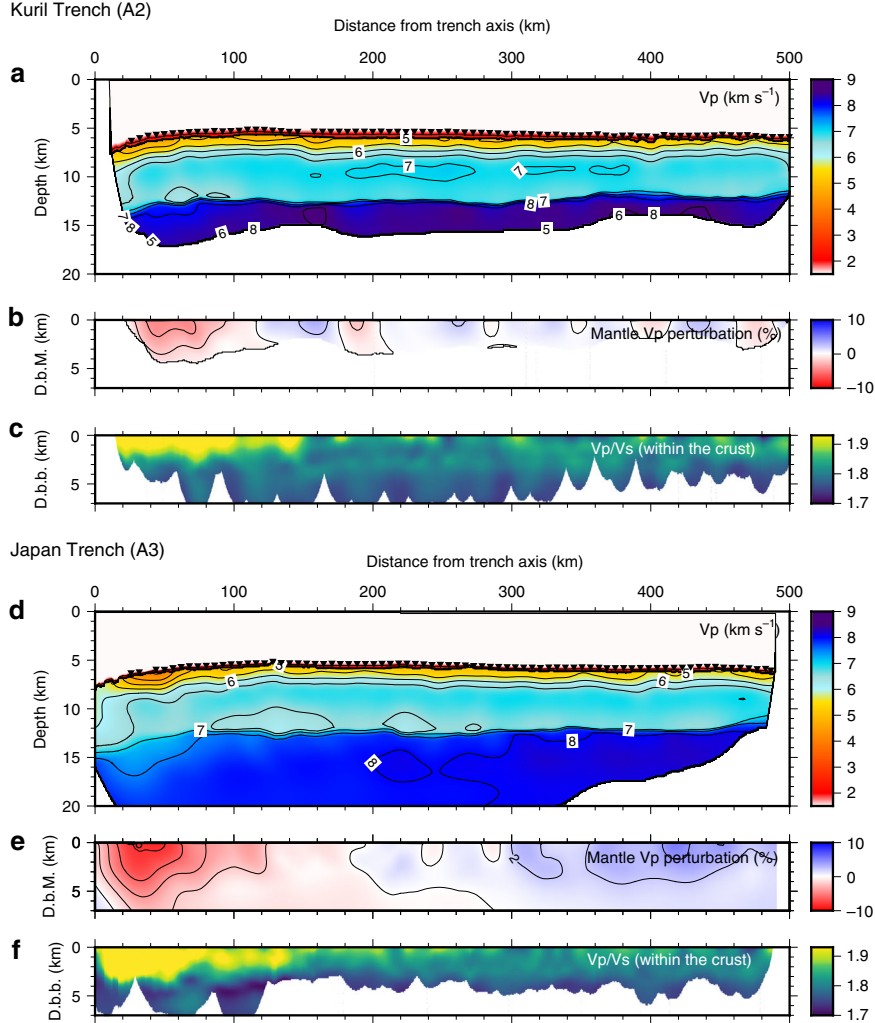

**Fig. 3** Seismic velocity models derived from traveltime data of controlled-source seismic surveys. **a–c** $V_p$, $V_p$ perturbation within the mantle, and the $V_p/V_s$ ratio within the crust along the Kuril Trench profile. **d–f** $V_p$, $V_p$ perturbation within the mantle, and the $V_p/V_s$ ratio within the crust along and the Japan Trench profile. The inverted triangles in **a** and **d** show OBS locations. D.b.M. is depth below the Moho, and D.b.b. is depth below the basement. To enable direct comparisons between the trenches, only the OBSs shown by green circles in Fig. 1 were used to develop these velocity models. Results obtained by using all OBSs are shown in Supplementary Fig. 6. The reduction of mantle $V_p$ become significant near the trench along both lines, which is well correlated with the development of normal faults (Supplementary Fig. 3b). The increase of $V_p/V_s$ ratio within the crust starts at about 150 km from the trench axis and are well correlated with the roughness (continuity) of the basement reflection (Supplementary Fig. 3c), suggesting fracturing at the basement promotes water penetration into the oceanic crust[10]

both trenches, but the magnitude of the changes was not the same. First, the $V_p$ reduction penetrates several times deeper in the Japan Trench than in the Kuril Trench; for example, a 2% $V_p$ reduction reaches 7 km below the Moho in the Japan Trench but only 2 km below in the Kuril Trench, suggesting that the volume of altered mantle is several times larger in the Japan Trench than in the Kuril Trench. Second, the reduction rate of mantle $V_p$ near the trench reaches ~12% in the Japan Trench, but no more than 8% in the Kuril Trench. Provided all of the mantle $V_p$ reduction is caused by serpentinization and that the reduction rate is not dependent on seismic anisotropy, the degree of serpentinization can be roughly estimated to be 25% (water content ~3wt%) in the Japan Trench, and at most 15–20% (water content ~2wt%) in the Kuril Trench[21,22]. If the $V_p$ reduction rate is considered in combination with the penetration depth in the mantle, mantle hydration is much larger in the Japan Trench than in the Kuril Trench. In addition, the thicker high-$V_p/V_s$ layer in the Japan Trench also supports the inference that more water penetrates

into the oceanic plate in the Japan Trench than in the Kuril Trench. Thus, our seismic velocity models indicate that plate hydration before subduction is remarkably higher in the Japan Trench than in the Kuril Trench.

The degree of plate hydration after subduction is often evaluated by the dehydration-induced intermediate-depth earthquakes activity occurring within the subducting oceanic plate[19,23]. In Japan, a dense nationwide seismic network has revealed the detailed distribution of intermediate-depth earthquakes in these subduction zones[24]. There are two seismic planes in this area and upper plane earthquakes occur mostly in the subducting oceanic crust and the topmost mantle. Upper plane seismic activity in the Japan Trench subduction zone is 5–6 times that in the Kuril Trench subduction zone (Fig. 2e), suggesting that the subducting oceanic plate is much more hydrated at the Japan Trench than at the Kuril Trench.

The good agreement between plate hydration before subduction, revealed by our seismic structure models, and plate

hydration after subduction, revealed by the frequency of dehydration-induced earthquakes, indicates that spatial variations in plate hydration are controlled primarily by the spatial variations in bend faulting in this subduction zone.

## Discussion

The differences in bend faulting between these two trenches are attributable to differences in the type of bend faulting. Generally, there are two types of bend faults: reactivated abyssal-hill faults and newly created faults[25,26]. When an oceanic plate is formed at an oceanic spreading ridge, abyssal-hill faults are created parallel to the spreading axis. At intermediate- and fast-spreading ridges, abyssal-hill faults reach depths 3–5 km below the seafloor[26] and they are spaced a few kilometers apart[27]. Ancient abyssal-hill faults are considered to be weak zones, and they are easily reactivated if the bending axis (i.e., the trench axis) is subparallel to the abyssal-hill fabric (i.e., the ancient spreading ridge axis). Compilations of the world's subduction trenches[25,26] show that abyssal-hill faults are reactivated when the trench-ridge angle $\alpha$ is <25–30°; otherwise bend faults are newly created that cross pre-existing abyssal-hill faults.

In the north-western Pacific margin (Fig. 1), the abyssal-hill fabric is subparallel to the Kuril Trench ($\alpha \sim 10°$), but oblique to the Japan Trench ($\alpha = 60$–70°). Thus, the bend faults at the Kuril Trench are reactivated abyssal-hill faults whereas those at the Japan Trench are newly created faults. On the outer slope of the Kuril Trench, the strike and spacing of the bend faults are consistent with those expected for reactivated abyssal-hill faults. On the other hand, at the Japan Trench, the bend faults are subparallel to the bending axis (trench axis) and intersect the abyssal-hill fabric; as a result, the fault network consists of both new faults and ancient faults. This network of damaged and weak zones might promote more water penetration into the oceanic plate than the isolated faults at the Kuril Trench, and may be a plausible explanation for the differences in plate hydration between these subduction zones.

Compared with the reactivation of abyssal-hill faults, the formation of new faults is considered to be more difficult. Thus, there will be a concentration of extensional stress around a newly formed fault. The spacing between newly formed faults in the Japan Trench is at least twice that between reactivated faults in the Kuril Trench (Fig. 2), indicating that the magnitude of the plate bending-related extensional stress concentrated around each bend fault in the Japan Trench is at least twice that around the reactivated faults in the Kuril Trench. This result is consistent with the differences in the cumulative displacement of the bend faults between these trenches. The stress concentration in the Japan Trench facilitates the occurrence of large, normal faulting earthquakes cutting across the entire oceanic crust, which are essential for mantle hydration. Thus, a newly formed fault promotes mantle hydration much more effectively and the type of bend faulting, that is, the trench-ridge angle $\alpha$, can therefore be considered a key controlling factor of plate hydration in the north-western Pacific margin.

A relationship between the trench-ridge angle $\alpha$ and plate hydration has been observed at the Alaska and the Middle American Trenches[19,28]. In both these subduction zones, the trench axes are nearly straight but the incoming oceanic plates, which have undergone reorganization, show remarkable along-trench segmentation. Plate hydration after subduction is larger along reactivated abyssal-hill fault segments ($\alpha < 25°$) than along segments where $\alpha > 30°$. These observations seemingly contradict findings in the north-western Pacific margin; however, the along-trench variations in plate hydration in these subduction zones can also be explained by the development of

bend faulting. Reactivated bend faults are clearly observed in the segments where $\alpha < 25°$, but almost no bend faulting can be recognized in the segments where $\alpha > 30°$; thus, in these subduction zones, plate hydration is dependent on the development of bend faulting. The lack of bend faults in segments where $\alpha > 30°$ in these subduction zones can be explained by (1) difficulties in forming new faults because the plate curvature is significantly smaller than that in the north-western Pacific margin, and (2) relatively younger plate ages which might influence the rigidity and flexibility of the oceanic plate. In addition, we cannot exclude the possibility that along-trench variations in plate hydration are related to pre-existing tectonic features, because in these subduction zones each plate segment has a very different tectonic history.

Our seismic surveys around the junction of the Japan and Kuril trenches have revealed that subducting plate hydration is strongly dependent on plate bending-related faulting prior to subduction. Differences in plate hydration can be explained by the type of bend faulting; plate hydration is more effective where the faults are newly formed than where they are reactivated abyssal-hill faults. The type of bend faulting is dependent on $\alpha$, the angle between the current trench axis and the ancient spreading ridge axis. Thus, we propose that the trench-ridge angle is a key controlling factor of the spatial variations in plate hydration in the north-western Pacific margin. However, spatial variations in plate hydration are also likely to be affected by other structural features such as plate curvature, plate age, and the presence of transform faults on the oceanic plate. Further, we need additional information, such as electric resistivity measurements, to quantitatively assess water transport by the oceanic plate. A broader range of studies will be essential to reveal the details of water transport by subducting oceanic plates.

## Methods

**Data acquisition.** We deployed 80 Ocean Bottom Seismometers (OBSs) along line A2 (Kuril Trench) and 86 OBSs along line A3 (Japan Trench) at a spacing of 6 km and fired a large airgun array (total volume 7800 cubic inches) from R/V Kairei at 200-m intervals for OBSs and at 50-m intervals for a 444-channel, 6-km-long hydrophone streamer. In OBS vertical component data, we observed good quality P-wave refractions from the oceanic crust and mantle, and good quality wide-angle reflections from the Moho discontinuity (Supplementary Figs 1, 2). In addition, we observed S-wave refractions and reflections in the radial (horizontal) component data. On time-migrated multi-channel seismic reflection sections, we could trace the basement (top of the oceanic crust) and the Moho clearly except in the areas with bend faults (Supplementary Fig. 3).

**Data processing.** We modeled $V_P$ and $V_S$ structures by a tomographic traveltime inversion method[10,29]. For $V_P$ modeling, we divided seismic structure models into four layers (seawater, sediment, crust, and mantle) separated by three interfaces (seafloor, basement, and Moho) and determined $V_P$ by using crustal and mantle refractions and Moho reflections in OBS data, and two-way traveltimes from the basement and Moho in multi-channel seismic data (Fig. 3, Supplementary Figs. 4, 5).

To model $V_S$ and the $V_P/V_S$ ratios, we utilized P-to-S conversion phases as well as P-waves. In our study area, the most efficient P-to-S conversion interface was considered to be the sediment basement. We therefore assumed that large-amplitude P-to-S phases were converted at the basement. Generally, there are two P-to-S conversions: P-to-S conversion of the ascending ray (PPS) and P-to-S conversion of the descending ray (PSS). The PPS is converted just beneath each OBS and has almost the same apparent velocity as P. The PSS is converted just beneath the airgun shot point and has the apparent S-wave velocity of the crust. The PPS can effectively constrain $V_S$ within the sediments, and the PSS can effectively constrain $V_S$ below the sediment basement.

Unlike P-wave mantle refractions, S-wave mantle refractions were not clearly observed in the horizontal data (radial component) near the trench axis where bend faults are well developed (Supplementary Figs 1a, 2a). Thus, we focused on the crustal structure and adopted three-layer model parameterization (seawater, sediment, and crust) to develop the $V_S$ and $V_P/V_S$ ratio models. We adopted the following 3-step inversion scheme to determine $V_S$ models. First, we determined the $V_P$ structure by tomographic traveltime inversion using first arrivals and multi-channel seismic two-way reflection traveltimes from the basement. Second, we modeled $V_S$ within the sediment by traveltime inversion using only the PPS phases.

Finally, we modelled $V_s$ beneath the basement by tomographic inversion using PSS phases. We calculated the $V_p/V_s$ ratio directly from these $V_p$ and $V_s$ models.

**Uncertainty of the obtained models.** Inversion results are generally dependent on the starting models as well as on the data. We evaluated the dependency by a Monte Carlo approach[29–31] as follows. First, we constructed a wide range of starting models randomly. Then, we applied tomographic traveltime inversion to all starting models to determine the final seismic velocity models. Finally, we calculated the average model and its standard deviation (SD). We considered the SD to be the uncertainty of the obtained seismic velocity models.

The random parameters for the starting $V_p$ models were $V_p$, $V_p$ gradient, and layer thickness. The starting model for $V_s$ inversion was generated from each final $V_p$ model by assuming a $V_p/V_s$ ratio of 8.0 in the sediment and 1.75 below the basement. In both $V_p$ and $V_s$ modeling, we generated 400 final models that fulfilled the condition $\chi^2 < 1$, where $\chi^2$ is the normalized sum of the root mean square of traveltime misfits divided by the uncertainty of each traveltime pick. Then, we calculated the average $V_p$ models (Fig. 3) and their SDs (Supplementary Figs 4-6).

The SD of the final $V_p$ and $V_s$ models within the oceanic upper crust and topmost mantle was usually less than 0.1 km s$^{-1}$ except at the two ends of the models. The SD of the final $V_p$ within the topmost mantle became a little larger near the trench axis, but it was still usually <0.2 km s$^{-1}$, indicating that the differences we found in $V_p$ reduction between the Kuril Trench and the Japan Trench were significant. On the other hand, we could not discuss the details of lower crustal $V_p$ because the SD of lower crust was relatively large. This is because the trade-off between the lower crustal $V_p$ and the Moho depth (crustal thickness) were not well constrained by our data.

In addition to the uncertainty analysis, we applied checkerboard resolution tests (CRT) to evaluate the spatial resolution of velocity perturbation (Supplementary Fig. 4-6). The checkerboard size is 30×3 km and we gave 5% perturbation. The checkerboard patterns were well recovered within the upper crust for both of $V_p$ and $V_s$ tomography and within the topmost mantle for $V_p$ tomography, which suggest that the lateral velocity perturbations there are reliable at the scale of 30 km. On the other hand, especially in the Japan Trench (Line A3), the checkerboard patterns could not be well recovered within the lower crust for $V_p$ tomography due to the lack of laterally propagating phases within the lower crust. Thus, we focus on the seismic velocity variations in the upper crust and the topmost mantle in this study.

**Bathymetry data.** Seafloor bathymetric data in Figs. 1 and 2 are ETOPO1, and a compilation of multi-narrow beam echo sounder data collected by Japan Coast Guard and JAMSTEC[32]. These figures were plotted using the Generi Mapping Tool software [33].

## Data availability

All the seismic survey data can be requested from JAMSTEC crustal structural database site (https://www.jamstec.go.jp/jamstec-e/IFREE_center/index-e.html).

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

## Acknowledgements

We gratefully acknowledge the captain, crew and marine technicians of R/V Kairei. This study was partly supported by a KAKENHI grant (15H05718) from the Japan Society for the Promotion of Science.

## Author contributions

G.F. and S.K. designed the seismic surveys. Y.K., Y.Y., G.F., and T.T., participated in the research cruises and collected seismic survey data. T.Y. provided ultra-deep OBSs and pre-processed the data. S.M. provided logisitcal support for seismic data acquisition and

processing. G.F. analyzed the seismic survey data and wrote the manuscript with contributions from all of the co-authors.

## Additional information

**Competing interests:** The authors declare no competing interests.

