## [Peer Review File · Nature Communications]

REVIEWERS' COMMENTS:

Reviewer #1 (Remarks to the Author):

Dear Editor,

In this revised manuscript, Fujie et al. report results from an active source seismic experiment across the Japan and Kuril trenches. Their tomography models utilize both reflection and refraction data to constrain the P- and S-wave velocity of the crust and uppermost mantle as a function of distance seaward of the trench. The primary novel conclusion of this work shows that the development of new bending faults on an oceanic plate where the abyssal hill fabric is oriented oblique to the trench axis yields larger sub-basement velocity reductions compared with reactivated faults where the fabric is parallel to the trench. This is important since prior studies have considered regions with reactivated bending faults where abyssal hill fabric and trench axis are parallel, or where abyssal hill fabric and trench axis are oblique with no evidence of bend fault development (e.g., Shillington et al. 2015). The new data are compelling and provide unique constraints that are likely of broad interest, and the revisions satisfy my concerns with the original manuscript. Therefore, I recommend this manuscript for publication.

One minor issue, the fifth panel in Fig. 3 (D.b.M. %perturbation of Japan Trench profile) now shows as blank and needs correction.

Best,
Samer Naif

Response to reviewer #2 (Dr. Samer Naif)

Reviewer comment 1:

One minor issue, the fifth panel in Fig. 3 (D.b.M. %perturbation of Japan Trench profile) now shows as blank and needs correction.

Response:

Thank you for the comment. We have remade Fig. 3 with different color palettes and confirmed that the PDF file of Fig. 3 is displayed as we intended on MS-Windows 10 (Adobe Reader and Microsoft Edge) and Linux (Evince, ghostscript, and Foxit PDF Reader).

We wish to thank the reviewer, Dr. Naif and an anonymous reviewer again for their comments, which helped improve this manuscript.